# Towards Low-Resource Automatic Program Repair with Meta-Learning and Pretrained Language Models

**Weishi Wang[12], Yue Wang[1*], Steven C.H. Hoi[1], Shafiq Joty[12]**
[1]Salesforce AI Research
[2]Nanyang Technological University, Singapore
{weishi.wang,wang.y,sjoty,shoi}@salesforce.com

## Abstract

Automatic program repair (APR) has gained increasing attention as an essential technique in software development to reduce manual debugging efforts and boost developers' productivity. Recent advances in deep learning (DL) based models have demonstrated promising results by learning from large-scale bug-fix examples in a data-driven manner. However, in practical scenarios, software bugs have an imbalanced distribution, and the fixing knowledge learned by APR models often only capture the patterns of frequent error types, making it inapplicable to handle the rare error types. To address this limitation, we investigate a novel task of low-resource APR, and propose Meta-APR, a new meta-learning framework integrated with code pretrained language models to generate fixes for low-resource bugs with limited training samples. Our Meta-APR learns better error-specific knowledge from high-resource bugs through efficient first-order meta-learning optimization, which allows for a faster adaptation to the target low-resource bugs. Besides, while we adopt CodeT5, a pretrained code-aware encoder-decoder Transformer, as the backbone model for Meta-APR, it is a model-agnostic framework that can be integrated with any neural models. Extensive experimental results on three benchmarks in various programming languages verify the superiority of our method over existing DL-based APR approaches.

## 1 Introduction

Program repair is critical to improving the productivity and stability of software development. However, it is resource-consuming and cost-prohibitive (Weiß et al., 2007; Planning, 2002; Jørgensen and Shepperd, 2007). A reliable automatic program repair (APR) system is thus crucial to reduce manual debugging efforts and development time (Gazzola et al., 2019; Winter et al., 2023).

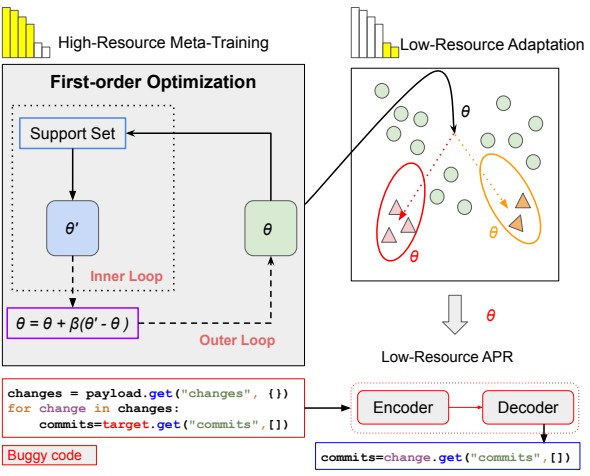

Figure 1: Illustration of our Meta-APR framework with CodeT5 for low-resource error-specific automatic program repair. We first meta-train the CodeT5 with high-resource bugs (●), where the backbone model is updated via gradient descent with respect to $\theta$. After that, Meta-APR is finetuned on the target low-resource bugs (▲,▲) using few-shot examples (10, 50, 100).

With the advances in deep learning (DL) models (Vaswani et al., 2017) and accessibility to large code corpora (Tufano et al., 2019; Lu et al., 2021), neural approaches to APR have achieved remarkable performance via exploiting existing code patches (Chen et al., 2021b; Zhu et al., 2021). These models are typically trained and evaluated on datasets that comprise a mix of various error types, which are diverse in nature: they vary in terms of the number of bug-fix pairs per error type, and are typically imbalanced. Moreover, the performance gaps across different error types are tremendous (Berabi et al., 2021), which can significantly impair the APR models' performance.

These observations motivate us to consider the idea: rather than training the model jointly on all error types, could we train a model that is quickly adaptable to any low-resource error-specific APR task? Inspired by the success of meta-learning on low-resource NLP tasks like machine transla-

---

*Corresponding author: wang.y@salesforce.com.

tion (Gu et al., 2018; Park et al., 2021) and dialogue generation (Mi et al., 2019; Lin et al., 2021), in this work, we drive pioneering efforts in formalizing low-resource APR, and propose an effective meta-learning framework that utilizes a code pretrained model to enhance APR performance.

**Low-resource APR formulation:** Unlike traditional APR approaches that jointly learn a model from a mix of error types, our formulation considers each rare error type as a low-resource target task. Accordingly, we create datasets specifically to support the evaluation of low-resource error-specific APR, based on three practical APR benchmarks in various programming languages: TFix in JavaScript (Berabi et al., 2021), ManySStuBs4J in Java (Karampatsis and Sutton, 2020), and TSSB-3M in Python (Richter and Wehrheim, 2022). We observe diverse and imbalanced error type distributions in these benchmarks, e.g., TFix, ManySStuBs4J, and TSSB-3M respectively have 31, 8, and 9 error types that are of low-resource[1] along with 21, 6, and 14 high-resource error types.

**Meta learning for low-resource APR:** To better address the task distribution issue while adapting the model to low-resource (synonymously, *few-shot*) tasks, we propose a novel meta-learning approach integrated with pretrained code language models. To the best of our knowledge, this is the first work to study the low-resource error-specific APR. We build Meta-APR with a code-aware pretrained encoder-decoder Transformer model CodeT5 (Wang et al., 2021) and an efficient first-order meta-learning algorithm Reptile (Nichol et al., 2018) for the challenging low-resource APR tasks. Fig. 1 illustrates the overview of our Meta-APR approach. Specifically, we first meta-train a CodeT5-based APR model on high-resource bug-fix pairs to learn a better model initialization that captures error-specific knowledge, which enables faster adaptation to the target low-resource bugs via finetuning on few-shot examples. In our experiments, we show that Meta-APR effectively aligns the representations between high-resource and low-resource bugs so that they have a closer distance in the representation vector space.

We extensively evaluate Meta-APR on three curated low-resource multilingual APR benchmarks with different degrees of low-resource settings, i.e. different numbers of training samples (10, 50, 100).

We show that Meta-APR significantly outperforms the standard transfer-learning method in all settings. As our Meta-APR is a model-agnostic framework that can be integrated with any other DL models, we compare its performance when integrated with other pretrained models like UniXcoder (Guo et al., 2022). Our results demonstrate that Meta-APR consistently enhances performance. Further analysis confirms that Meta-APR is a more robust and effective approach in fixing bugs with various buggy patch lengths and error types.

We further compare with closed-sourced language models such as ChatGPT (OpenAI) in fixing these low-resource bugs. We find Meta-APR achieves much better performance than ChatGPT under zero-shot/few-shot settings. Besides, we observe that ChatGPT often predicts "no bugs", which is probably because it does not well capture the fixing patterns of these low-resource bugs due to their data scarcity issue.

## 2 Related Work

**Automatic Program Repair (APR)** Recently, there is a growing body of APR research that aims to automate the rectification of software defects with less human intervention. In general, conventional APR approaches can be divided into three categories (Zhang et al., 2023, 2022), which are 1) *heuristic-based* (Goues et al., 2012; Qi et al., 2014; Jiang et al., 2018), 2) *constraint-based* (Martinez and Monperrus, 2018; Nguyen et al., 2013; Xuan et al., 2017), 3) *template-based approaches* (Liu et al., 2019; Koyuncu et al., 2020).

Besides, *learning-based approaches* (Chen et al., 2021b; Lutellier et al., 2020; Jiang et al., 2021) have shown to achieve promising results by learning the fix patterns from previous bug-fix pairs in an end-to-end manner. Motivated by the success of Neural Machine Translation (NMT) in the NLP domain, one notable learning-based APR method is formulated as a sequence-to-sequence generation problem (Tufano et al., 2019), which aims to translate a buggy code into its correct version. This technique is further enhanced by using pretrained models such as T5 (Raffel et al., 2020) in Berabi et al. (2021). In this work, we propose to exploit a pretrained code-aware CodeT5 (Wang et al., 2021) following Bui et al. (2022); Wang et al. (2023a).

**Meta-Learning for Low-Resource Tasks** Meta-learning has been well studied for few-shot learning as a learning-to-learn approach, which attempts to

---

[1]We select low-resource scenarios based on the number of examples per error type for each dataset.

learn new concepts based on past experiences (Bengio et al., 2013; Vilalta and Drissi, 2002). Recently, *optimization-based techniques* yield substantial improvement in many low-resource NLP tasks (Zhao et al., 2022). Among them, Model-Agnostic Meta-Learning (MAML) (Finn et al., 2017) has been widely used to tackle low-resource NLP tasks such as machine translation (Gu et al., 2018; Park et al., 2021), dialogue generation (Mi et al., 2019; Lin et al., 2021), and text-to-speech alignment (Lux and Vu, 2022). MAML has shown exceptional efficacy in learning a good parameter initialization for a fast adaption with limited resources.

Recently, meta-learning approaches have been adapted to solve low-resource software intelligence tasks such as code summarization (Rauf et al., 2022; Xie et al., 2022), and code search (Chai et al., 2022). To the best of our knowledge, we are the first to formulate the low-resource error-specific APR task based on the error type distributions and investigate the effectiveness of meta-learning methods. In addition, unlike prior approaches that mostly use the second-order meta-learning algorithm MAML, we exploit a more efficient first-order meta-learning method Reptile (Nichol et al., 2018). In the ablation studies, we show that it outperforms MAML in various low-resource settings.

**Programming Language Models**   Inspired by the success of pretrained language models (LMs) such as BERT (Devlin et al., 2019), GPT (Radford et al., 2018), and T5 (Raffel et al., 2020) in NLP tasks, there are many recent attempts for code pretrained models that can be classified as three categories: 1) Encoder-only approaches like CodeBERT (Feng et al., 2020) and GraphCode-BERT (Guo et al., 2021); 2) Decoder-only methods such as CodeGPT (Lu et al., 2021); and 3) Encoder-decoder models like CodeT5 (Wang et al., 2021; Le et al., 2022; Wang et al., 2023b). Besides, UniX-coder (Guo et al., 2022) adopts a UniLM-like architecture (Dong et al., 2019) with various attention masks. Recent studies further explore the use of very large LMs such as Codex (Chen et al., 2021a) for APR tasks (Prenner and Robbes, 2021; Joshi et al., 2022) in a zero-shot/few-shot setting, where there is still a clear gap between Codex and the domain-specific finetuning methods.

## 3   Approach

Fig. 1 illustrates the overview of our proposed Meta-APR, a meta-learning framework that lever-

---

**Algorithm 1:** Meta-Training for APR

**Require:** A set of high-resource error types
$\mathcal{T}_h = \{\mathcal{T}_1, \mathcal{T}_2, \ldots, \mathcal{T}_n\}$, $\forall \mathcal{T}_i \in \mathcal{T}_h$ it pairs
with associated bug-fix pairs that
$\mathcal{D}_i = \{(B_j, F_j)\}_{j=1}^{|\mathcal{D}_i|}$, a APR model $f_\theta$,
inner loop learning rate $\alpha$, outer loop
learning rate $\beta$, meta update step size $\mathcal{M}$

**Initialize:** Initialize $\theta$ from the APR model $f_\theta$
**Output:** Optimal meta-trained APR model $f_\theta$
**while** *not done* **do**
    $\mathcal{D}_h = \emptyset$.
    **forall** $\mathcal{T} \in \mathcal{T}_h$ **do**
        | Append the training dataset of $\mathcal{T}$ into $\mathcal{D}_h$
    **end**
    Randomly divide the merged training dataset $\mathcal{D}_h$
      into batches $\mathcal{B}_s$
    **forall** $\mathcal{B}_s$ **do**
        Obtain $\mathcal{B}_s^{support}$ and $\mathcal{B}_s^{querry}$ as in §3.1
        Evaluate the inner loop cross entropy loss
          $\mathcal{L}_{inner}(f_\theta, \mathcal{B}_s^{support})$
        Update error-specific model parameters with
          gradient descent:
        $\theta' = \theta - \alpha \nabla_\theta \mathcal{L}_{inner}(f_\theta, \mathcal{B}_s^{support})$
        **if** *current step $i$* $\mod \mathcal{M} = 0$ **then**
          | Update global model parameters with
            estimation: $\theta \leftarrow \theta + \beta(\theta' - \theta)$
        **end**
        Evaluate the inner loop cross entropy loss
          $\mathcal{L}_{inner}(f_\theta, \mathcal{B}_s^{querry})$
    **end**
**end**

---

ages a code pretrained model for low-resource error-specific APR. We first formulate the task of low-resource error-specific APR in §3.1. Then, we describe our error-specific meta-APR dataset creation in §3.2 and our Meta-APR method in §3.3.

### 3.1   Task Formulation

Assume that we have a set of error types $\mathbb{T} = \{\mathcal{T}_1, \mathcal{T}_2, \ldots, \mathcal{T}_n\}$. For each error type $\mathcal{T}_i$, it associates with a collection of bug-fix pairs $\mathcal{D}_i = \{(B_j, F_j)\}_{j=1}^{|\mathcal{D}_i|}$, where $(B_j, F_j)$ denotes the $j$-th bug-fix pair. For the error types in $\mathbb{T}$, we define their resourceness based on the total number of bug-fix pairs $|\mathcal{D}_i|$. Considering the actual data distribution across three benchmarks, we select an empirical cutoff value of 1000 instances. This threshold value is established to identify an error type as low-resource if it has less than 1000 samples. Otherwise, we treat it as high-resource.

Formally, our proposed framework comprises a neural sequence-to-sequence (seq2seq) model (Sutskever et al., 2014) $f_\theta$ as a base APR model. Given a set of error-specific bug-fix pairs $\mathcal{D}_i = \{(B_j, F_j)\}_{j=1}^{|\mathcal{D}_i|}$, $f_\theta$ generates $F_j$ based on $B_j$

| ET | no-extra-bind | SWAP_BOOLEAN_LITERAL | SAME_FUNCTION_WRONG_CALLER |
|---|---|---|---|
| **Patch difference** | ```return mapCb(err, memo);`
`    -}.bind(this)`
`    -}.bind(this), waterCb;`
`    +});`
`    +}, waterCb;`
`}.bind(this),``` | ```try (LockedInodePath inodePath =`
`mInodeTree.lockFullInodePath`
`      (entry.getId(),`
`InodeTree.LockMode.WRITE))`
`{`
`  - setAttributeInternal(inodePath,`
`false, entry.getOpTimeMs(), options);`
`  + setAttributeInternal(inodePath,`
`true, entry.getOpTimeMs(), options);`
`}}``` | ```class PushEventHook(BaseEventHook):`
`  changes =`
`self.payload.get("push",`
`{}).get('changes', [])`
`  for change in filter(None,`
`changes):`
`    - commits =`
`target.get("commits", [])`
`    + commits =`
`change.get("commits", [])`
`      if not commits:`
`        continue``` |
| **TF** | ```return mapCb(err, memo);`
`    }.bind(this)`
`    }.bind(this), waterCb;`
`}.bind(this),``` | ```setAttributeInternal(inodePath,`
`false, entry.getOpTimeMs(), options);``` | ```commits = target.get("commits", [])``` |
| **Meta-APR** | ```return mapCb(err, memo);`
`    });`
`    }, waterCb);`
`}.bind(this),``` | ```setAttributeInternal(inodePath, true,`
`entry.getOpTimeMs(), options);``` | ```commits = change.get("commits", [])``` |
| | (a) TFix (JavaScript) | (b) ManySStuBs4J (Java) | (c) TSSB-3M (Python) |

Figure 2: Bug fix examples on three low-resource error-specific APR tasks from one particular error type (ET), where our Meta-APR successfully fixes bugs while the transfer-learning (TF) approach fails to do so.

in an autoregressive manner. Formally,

$$P(F_j|B_j, f_\theta) = \prod_{k=1}^{N} P(F_{j,k}|B_j, F_{j,1} : F_{j,k-1}, f_\theta)$$

where $F_{j,1} : F_{j,k-1}$ is the previous sequence at $k$-th token with $N$ denoting the total number of tokens in the target sequence $F_j$.

During the meta-training stage, we randomly sample a batch of bug-fix pairs $\mathcal{B}_s = \{(B_s, F_s)\}_{s=1}^{|\mathcal{B}_s|}$ from high-resource error types. Each batch $\mathcal{B}_s$ is further divided into $\mathcal{B}_s^{support}$ and $\mathcal{B}_s^{query}$ equally. Then, we apply the first-order meta-learning algorithm Reptile (Nichol et al., 2018) to update $f_\theta$ via gradient descent. After that, the model $f_\theta$ is finetuned on a low-resource error type with few-shot examples. The underlying idea of Meta-APR is to meta-train a model on high-resource error types such that it is quickly adaptable to low-resource types with few-shot examples.

## 3.2 Low-resource APR Dataset Construction

As there are no available low-resource APR benchmarks for evaluation, we curate three low-resource APR datasets in various low-resource settings from three existing APR benchmarks with error type annotations, which are TFix in JavaScript (Berabi et al., 2021), ManySStuBs4J in Java (Karampatsis and Sutton, 2020), and TSSB-3M in Python (Richter and Wehrheim, 2022). As mentioned in §3.1, we define the low-resource error types

based on the actual counts of its associated bug-fix pairs ($< 1000$). To construct more challenging low-resource scenarios, we randomly select 10, 50, and 100 samples from each low-resource error type. Following the common practice (Gao et al., 2021), we repeat this few-shot sampling process with five different random seeds (13, 21, 42, 87, and 100). For evaluation, we report the averaged results over the five seeds to rule out the random noises.

## 3.3 Model-Agnostic Meta-APR Framework

**Base APR Model** CodeT5 (Wang et al., 2021) is a unified code-aware encoder-decoder Transformer model pretrained from large-scale source code corpus in eight different programming languages. CodeT5 has been shown to achieve SoTA performance in many code understanding and generation tasks such as defect detection and program refinement. In this work, we propose to adapt CodeT5 as the base model of our Meta-APR to leverage its better code understanding capability.

**High-Resource APR Meta-Training** During the meta-training phase, each mini-batch of data simulates the low-resource scenarios. In our Meta-APR approach, we iterate through a set of high-resource error types as a private training task to update $f_\theta$. We first merge all high-resource error-specific training dataset as $\mathcal{D}_h^{train}$, and randomly segment $\mathcal{D}_h^{train}$ into $N$ batches $\{\mathcal{B}_1, \mathcal{B}_2, \ldots, \mathcal{B}_N\}$ equally. Then, each $\mathcal{B}_s$ is further split into $\mathcal{B}_s^{support}$ and $\mathcal{B}_s^{query}$ to form a local error-specific meta-learning

task to update the global APR model $f_\theta$ using gradient descent:

$$\theta' = \theta - \alpha\nabla_\theta\mathcal{L}(f_\theta, \mathcal{B}_s^{support})$$
$$\theta \leftarrow \theta + \beta(\theta' - \theta)$$

where $\theta$ is the global model parameters, and $\theta'$ is the local error-specific model parameters, $\alpha$ and $\beta$ denote the learning rate of the inner loop and outer loop respectively, $\mathcal{L}$ denotes the cross entropy loss function. The error-specific local gradients are grouped by every $\mathcal{M}$ steps to update the global APR model parameters $\theta$. The meta-training procedure of our Meta-APR is summarized in Algorithm 1. In the low-resource setting, we set the size of support and query sets to 10 and we leverage support sets for the inner loop update. The query sets are used to track the meta-loss and not involved in parameter updating.

**Low-Resource APR Adaptation**   After the meta-training, we adapt Meta-APR to the target low-resource APR tasks via directly finetuning the meta-learned global APR model on few-shot training samples. Such meta-learned APR model is expected to capture error-specific knowledge by providing a better model initialization, which enables faster adaptation to fix low-resource bugs. In finetuning, the objective is to minimize the cross-entropy loss between model predictions and ground-truth fixes.

## 4   Experimental Setup

### 4.1   Error-Specific APR Dataset

**ManySStuBs4J (Karampatsis and Sutton, 2020)** has small and large versions comprising 10,231 and 63,923 bug-fix pairs respectively in Java. It is organized at the level of the single statement changes for each bug-fix pair. We consider ManySStuBs4J large with 14 error types in this work.

**TFix (Berabi et al., 2021)**   is a large-scale program repair dataset that consists of a ground truth repair code patch for each buggy patch in JavaScript. It focuses on syntax and stylistic errors from open-source GitHub commits, which comprise 104,804 bug-fix pairs. Among them, 52 error types are detected by a static analyzer ESLint [2] (Tómasdóttir et al., 2020).

[2] https://eslint.org/

| Benchmark | High-resource | | Low-resource | | |
|---|---|---|---|---|---|
| | #Error | #Train | #Error | Few-shots | #Test |
| ManySStuBs4J | 6 | 20,225 | 8 | (0,10,50,100) | 569 |
| TFix | 21 | 75,998 | 31 | (0,10,50,100) | 1,087 |
| TSSB-3M | 14 | 66,384 | 9 | (0,10,50,100) | 538 |

Table 1: Data statistics of 3 error-specific low-resource APR benchmarks. During low-resource finetuning, we randomly sample (10,50,100) shots for each error type to construct various low-resource settings.

**TSSB-3M (Richter and Wehrheim, 2022)**   is a dataset of over 3 million isolated single statement bug fixes across 23 error types. Each bug fix is associated with a commit in an open-sourced Python project that does not modify source code in other files or statements. We randomly down-sample by 10% for each error type.

To facilitate future research in this new field, we release our curated error-specific low-resource APR datasets at https://github.com/wang-weishi/Meta-APR. See Appendix A.1 for more detailed statistics.

**Data Preprocessing**   As discussed in §3.2, we process all three benchmarks to create high-resource and low-resource APR tasks based on the number of bug-fix in each error type. The data statistics are reported in Table 1. We further provide bug-fix examples for each benchmark in Fig. 2. To prepare the source input to Meta-APR, we follow Berabi et al. (2021) to combine error type, error message, and error context into a single piece of text in the following format:

**fix {error type} {error message} {error context}**

where error context consists of the given localized error line and its two neighboring code lines to form a buggy code patch. The corresponding fixed line is used as the target sequence.

### 4.2   Metrics and Baselines

**Metrics**   Following the common practice (Berabi et al., 2021), we use the Exact Match (EM) accuracy to measure the APR performance. Specifically, EM requires the prediction to be identical to the ground-truth fix, which can reflect how well model predictions are aligned with historic correct fixes from human developers. EM is commonly utilized to uphold correctness standards, especially in cases where static analyzers or unit tests are not available.

**Baselines** We compare Meta-APR with three learning settings: 1) only finetuning on low-resource bugs; 2) transfer-learning from high-resource to low-resource bugs; and 3) multi-task learning on both high-resource and low-resource bugs with or without upsampling strategies. Specifically, for the transfer-learning baseline, we first finetune the model on the high-resource training data and then have another stage of finetuning on the low-resource training data. Under the multi-task learning setting, we jointly finetune our models on a mix of both high-resource and low-resource training data.

Besides, we compare with other code pre-trained models as the backbone model, which include encoder-only CodeBERT (Feng et al., 2020), decoder-only UniXcoder (Guo et al., 2022), and encoder-decoder CodeT5 (Wang et al., 2021). For our Meta-APR method, we perform two ablation studies by replacing either the Reptile meta-learning approach to MAML or replacing the backbone model CodeT5 into UniXcoder to verify the effectiveness of our design choices.

### 4.3 Implementation Details

We implement Meta-APR based on the deep learning framework PyTorch[3]. We employ CodeT5-base[4] with 220M parameters as our backbone model. All of our experiments are conducted on a single NVIDIA A100-40GB GPU. We use the library Higher (Grefenstette et al., 2019) to meta-train the model on high-resource error types for 50 epochs with a batch size of 10, where the first 5 instances work as the support set and the remaining 5 instances are query set. For inner loop gradient updates, we use the SGD optimizer with an inner loop learning rate $\alpha$ of 1e-4. For the global gradient updates, we use the AdamW (Loshchilov and Hutter, 2019) optimizer and set the outer loop learning rate $\beta$ to 5e-5. Moreover, in the meta-training stage, we warm up the first 1000 steps with a linear decay. The meta update step size $\mathcal{M}$ is set to 150, 20, 150 for TFix, ManySStuBs4J, and TSSB-3M respectively. For low-resource APR adaptation, we finetune the meta-trained model for 50 epochs on low-resource error types with a batch size of 25 and a learning rate of 5e-5. For testing, we select the checkpoint which has the best EM on a held-out validation set.

---

[3] https://pytorch.org/
[4] https://github.com/salesforce/CodeT5/

| Method | Shot = 100 | Shot = 50 | Shot = 10 | Shot = 0 |
|---|---|---|---|---|
| *Low-resource finetuning* | | | | |
| CodeBERT | 6.43 | 3.64 | 0.14 | 0.00 |
| UniXcoder | 42.28 | 33.25 | 18.24 | 0.00 |
| CodeT5 | 42.74 | 36.10 | 9.24 | 0.00 |
| *Transfer-learning* | | | | |
| CodeBERT | 43.34 | 37.08 | 34.02 | 15.82 |
| UniXcoder | 53.43 | 47.66 | 34.87 | 18.10 |
| CodeT5 | 55.22 | 49.91 | 38.49 | 18.28 |
| *Multi-task learning* | | | | |
| CodeT5 | 53.78 | 46.75 | 35.85 | - |
| CodeT5 + upsampling | 58.98 | 52.73 | 41.09 | - |
| *Meta-learning* | | | | |
| Meta-APR | **59.44** | **54.34** | **42.04** | 22.50 |
| →MAML | 58.10 | 53.00 | 41.69 | **23.02** |
| →UniXcoder | 54.48 | 48.22 | 36.77 | 19.68 |

Table 2: Results on low-resource ManySStuBs4J.

| Method | Shot = 100 | Shot = 50 | Shot = 10 | Shot = 0 |
|---|---|---|---|---|
| *Low-resource finetuning* | | | | |
| CodeBERT | 4.91 | 3.05 | 0.00 | 0.00 |
| UniXcoder | 33.79 | 28.14 | 16.39 | 0.00 |
| CodeT5 | 35.61 | 29.07 | 13.05 | 0.00 |
| *Transfer-learning* | | | | |
| CodeBERT | 26.06 | 21.89 | 10.89 | 3.35 |
| UniXcoder | 40.15 | 35.28 | 24.98 | 15.80 |
| CodeT5 | 46.91 | 43.05 | 31.23 | 13.57 |
| *Multi-task learning* | | | | |
| CodeT5 | 46.47 | 41.26 | 30.30 | - |
| CodeT5 + upsampling | 46.84 | 41.38 | 29.41 | - |
| *Meta-learning* | | | | |
| Meta-APR | **47.77** | **44.83** | **35.28** | **24.72** |
| →MAML | 47.03 | 44.09 | 34.95 | 20.82 |
| →UniXcoder | 40.85 | 35.58 | 25.58 | 15.61 |

Table 3: Results on low-resource TSSB-3M.

## 5 Experimental Results and Analysis

In this section, we compare Meta-APR with other code pretrained models in different training settings on a set of our curated low-resource error-specific APR tasks from three benchmarks (§5.1), followed by a detailed analysis on the effects of different error types and token length (§5.2), and a pilot study to compare with a closed-sourced large language model such as ChatGPT in fixing these challenging low-resource bugs (§5.3).

### 5.1 Low-Resource APR Performance

Tables 2 to 4 present the results of exact match (EM) accuracies on ManySStuBs4J, TSSB-3M, and TFix benchmarks respectively at different low-resource settings. We can observe that Meta-APR consistently outperforms other baselines in various few-shot settings across 3 benchmarks in different programming languages. Among different models, we find that CodeT5 achieves consistent performance gains over CodeBERT and UniXcoder in most cases, demonstrating that it can serve as

| Method | Shot = 100 | Shot = 50 | Shot = 10 | Shot = 0 |
|---|---|---|---|---|
| *Low-resource finetuning* | | | | |
| CodeBERT | 13.15 | 1.75 | 0.13 | 0.00 |
| UniXcoder | 45.11 | 41.53 | 27.51 | 0.00 |
| CodeT5 | 45.85 | 40.18 | 24.58 | 0.09 |
| *Transfer-learning* | | | | |
| CodeBERT | 42.80 | 38.86 | 26.66 | 16.38 |
| UniXcoder | 46.64 | 44.89 | 32.75 | 18.31 |
| CodeT5 | 51.02 | 46.46 | 34.26 | 21.44 |
| *Multi-task learning* | | | | |
| CodeT5 | 50.60 | 47.29 | 34.41 | - |
| CodeT5 + upsampling | 51.39 | 47.58 | 36.28 | - |
| *Meta-learning* | | | | |
| Meta-APR | **51.63** | **48.06** | **39.50** | **24.38** |
| →MAML | 50.56 | 47.38 | 37.09 | 21.71 |
| →UniXcoder | 47.45 | 44.98 | 34.37 | 17.66 |

Table 4: Results on low-resource TFix.

a better backbone model for APR tasks with an encoder-decoder architecture.

Among different learning paradigms, we find that transfer-learning from high-resource to low-resource bugs and multi-task learning on both bugs yield much better results compared to directly finetuning on low-resource bugs, validating our assumption that low-resource APR can benefit from the bug-fixing knowledge learned from high-resource bug-fix data. These two approaches generally exhibit comparable performance across different benchmarks, and the upsampling strategy often proves to be helpful in multi-task learning. Overall, our Meta-APR further improves the adaptation from high-resource to low-resource bugs, thereby leading to superior APR performance. Notably, the performance gain of Meta-APR over other learning paradigms becomes more significant when there are fewer or even no low-resource training samples available. This implies that Meta-APR is able to learn a better model initialization that captures the error-specific knowledge, thereby enabling faster adaptation to the target low-resource error types.

**Ablation Study** We consider two variants of Meta-APR to verify the design choices in our proposed framework, where "→MAML" means that we replace the first-order meta-learning algorithm with a second-order meta-learning approach MAML, and "→UniXcoder" means that we change the backbone model CodeT5 to UniXcoder. From the results, we find that both CodeT5 and the first-order meta-learning algorithm are important in enhancing low-resource APR performance, observed by a consistent performance drop from these two variants in most settings across 3 benchmarks. Note that our Meta-APR's first-order meta-learning is

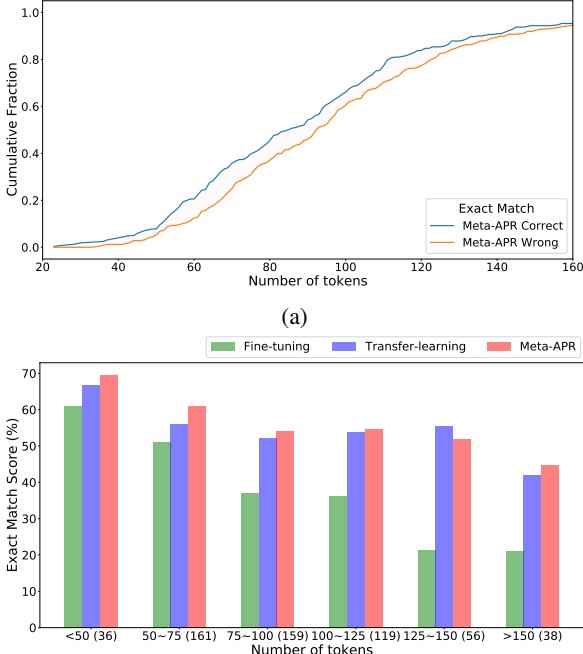

(a)

(b)

Figure 3: (a): cumulative fraction of programs by number of tokens in the source buggy patch, grouped by whether Meta-APR can have a correct fix. (b): distribution of correct fix over number of tokens for low-resource finetuning and transfer-learning from high-resource to low-resource bugs and our Meta-APR.

also more efficient than MAML's second-order meta-learning approach.

## 5.2 Further Analysis

We proceed to analyze the model predictions to better understand our Meta-APR behaves in fixing various bugs compared to other approaches. All results in this section are under a 100-shot setting.

**Effect of Bug Sequence Length** We analyze how Meta-APR performs in fixing low-resource bugs with varying numbers of bug tokens. Fig. 3a shows the cumulative fractions of bugs by their number of tokens, grouped based on the Meta-APR repair outcome (EM). Comparing the blue and orange lines, we observe that the blue one is consistently above the orange one, and if we select a fixed cumulative fraction based on the y-axis, the blue line (correct fixes) will have fewer tokens (i.e. shorter) than the orange one (wrong fixes), indicating the bugs successfully fixed by Meta-APR tend to be shorter than the ones that are incorrectly fixed. We further compare Meta-APR with other training strategies, based on the same backbone model of CodeT5, in fixing bugs with various lengths in Fig. 3b. We

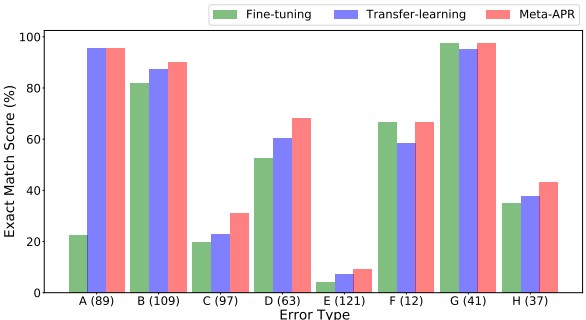

Figure 4: Distribution of correct fix on 9 low-resource error types from ManySStuBs4J. Number of bugs for each type is included in the parentheses at the x-axis. The details of error type A-H can be found in Table 5.

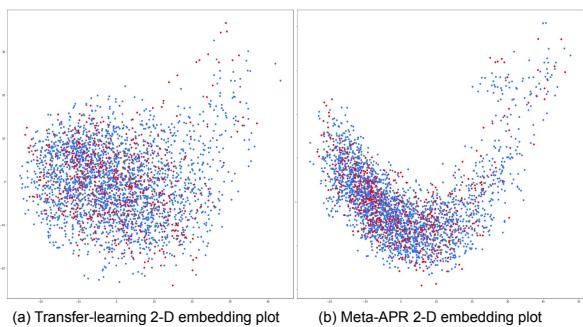

(a) Transfer-learning 2-D embedding plot   (b) Meta-APR 2-D embedding plot

Figure 5: 2-D visualization of embeddings for high-resource bugs (blue) and low-resource bugs (red).

observe a monotonous performance decline when the bug sequence length increases for all models, suggesting that shorter bugs are easier to be fixed which might be due to their limited complexity.

**Effect of Error Type** We further analyze how Meta-APR performs in addressing various error types. Fig. 4 presents a breakdown of results by error type for ManySStuBs4J, where we can find a notable variance in performance across different error types. For instance, Meta-APR achieves more than 50% EM score on types A, C, and E, while it achieves around 10% more EM score on type B. Overall, we observe that Meta-APR is a more robust APR method, consistently outperforming other finetuning strategies. Interestingly, finetuning only on low-resource bugs achieves comparable performance to Meta-APR in fixing bug type F (*add throws exception*) and G (*delete throws exception*). These bugs relate to the decision to add or remove a 'throws' clause in a function declaration, implying that such error types comprise easy-to-fix bugs and require only a few training samples.

**Representation Visualization** To understand how Meta-APR learns a better model initializa-

tion through error-specific meta-training compared to the default transfer-learning approach, we visualize the embeddings of both high-resource and low-resource bugs after the high-resource finetuning from transfer-learning and Meta-APR in Fig. 5. We observe that Meta-APR can better align the representations between high-resource and low-resource bugs so that they are distributed in a closer distance in the embedding vector space, enabling faster adaptation from high-resource to low-resource bugs with limited training samples.

**Case Study** We provide 3 qualitative examples from our multilingual low-resource APR benchmarks in Fig. 2. We find that our Meta-APR is able to fix the bugs using various fix operations such as deletions, boolean conversion, and identifier renaming, while the standard transfer-learning approach fails to fix bugs by simply copying the buggy line as the fixed line. This indicates Meta-APR can enable faster and better adaptation to low-resource APR scenarios.

## 5.3 Comparison with ChatGPT

Recent studies (Prenner and Robbes, 2021; Joshi et al., 2022) have shown that large language models (LLMs) are capable of bug fixing in zero-shot/few-shot settings. In order to investigate their performance in fixing challenging low-resource bugs, we use ChatGPT (GPT-3.5-Turbo[5]) and evaluate it on 80 randomly sampled test bug-fix pairs for each benchmark. As illustrated in Fig. 6, we construct the zero-shot prompt to provide the code context and its buggy line, together with an instruction "*fix the buggy line:*". Besides, we randomly select one bug-fix pair from the same error type to design the one-shot prompt for in-context learning.

We report the comparison results in Fig. 7. We observe that Meta-APR significantly surpasses ChatGPT in both zero-shot/one-shot settings across 3 tasks. This shows that ChatGPT is still lim-

---

[5] https://chat.openai.com/chat

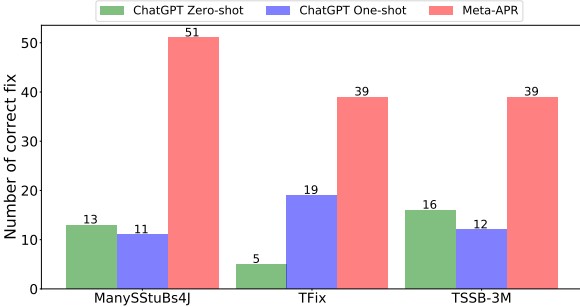

Figure 7: Evaluation results of correct fixes on a subset of 80 bugs from the test data across three benchmarks.

ited to handle the challenging low-resource bugs as it did not see much such bug-fix data during training due to the data scarcity issue. Additionally, we find that the one-shot example is not always beneficial for low-resource APR and might introduce some noises compared to zero-shot setting. It substantially improves the performance on TFix but leads to some performance degrades on ManySStuBs4J and TSSB-3M. By inspecting the predictions, we find that ChatGPT often predicts "no bugs" as it might require more semantic information for decision-making. Besides, ChatGPT performs pretty well in fixing bugs related to syntax errors such as the error type of "*no unsafe negation*", which is to fix the bug by simply adding parentheses to an expression after a negation operator. This is probably due to the fact that ChatGPT has been pretrained on a large-scale code corpus and can understand the program syntax well.

## 6 Conclusion

In this work, we present Meta-APR, a simple yet effective framework that extends CodeT5 with meta-learning for low-resource APR. It is a model-agnostic framework that can be integrated with any learning-based models. To the best of our knowledge, we are the first to investigate APR in the low-resource setting and curate error-specific datasets in different low-resource degrees from three APR benchmarks in Python, Java, and JavaScript. Comprehensive experiments have verified the superiority of Meta-APR over other learning strategies with various code pretrained models. More analysis shows that Meta-APR can better align the representations of high-resource and low-resource bugs, and fix bugs with various sequence lengths and error types. A pilot comparison with ChatGPT further shows that our Meta-APR is still more capable of fixing these challenging low-resource bugs.

## Limitations

As we are the first to investigate the low-resource APR tasks, we curated 3 datasets with different low-resource degrees (i.e., shot=10/50/100) from existing APR benchmarks to support our study. Such data construction will have a data quality dependency issue from those original APR datasets. Besides, the low-resource sub-sampling may introduce some randomness issues. To mitigate this issue, we performed multiple rounds of random sampling with different seeds and reported the average results. Furthermore, to evaluate the APR performance, we employ exact match scores as the metric to compare the predicted fixes with the ground-truth fixes written by developers, which might fail to capture other correct fixes with different formats and styles.

## Ethics Statement

Our work complies with ACL Ethics Policy. In this work, we construct our datasets using publicly available APR benchmarks, which are widely used to examine the program repair performance. We provide detailed procedures to create our low-resource APR datasets and provide proper citations to their source benchmarks. We will publicly release our curated datasets with the same licenses as their source datasets. As an APR tool, one potential risk of Meta-APR is that the predicted fixes from Meta-APR cannot be guaranteed to be correct, and directly adopting them without manual checking could cause security risks to the software development. We suggest that all the fixes should have a manual check from experts before real adoption.

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

# A  Appendix

## A.1  More Dataset Statistics

We provide the detailed statistics of our curated low-resource APR benchmarks in Table 5, Table 6, and Table 7 for ManySStuBs4J, TSSB, and TFix respectively. We can observe a very imbalanced error type distribution across these benchmarks.

| | Error Type | Train | Valid | Test | All |
|---|---|---|---|---|---|
| High-resource | CHANGE_IDENTIFIER | 10,175 | 1,250 | 1,250 | 12,675 |
| | OVERLOAD_METHOD_MORE_ARGS | 2,734 | 338 | 339 | 3,411 |
| | CHANGE_NUMERAL | 2,694 | 336 | 336 | 3,366 |
| | CHANGE_MODIFIER | 2,580 | 322 | 322 | 3,224 |
| | MORE_SPECIFIC_IF | 1,028 | 128 | 128 | 1,284 |
| | CHANGE_OPERATOR | 1,014 | 126 | 127 | 1,267 |
| Low-resource | LESS_SPECIFIC_IF (E) | 968 | 121 | 121 | 1,210 |
| | SWAP_BOOLEAN_LITERAL (B) | 871 | 109 | 109 | 1,089 |
| | OVERLOAD_METHOD_DELETED_ARGS (C) | 790 | 98 | 97 | 985 |
| | CHANGE_CALLER_IN_FUNCTION_CALL (A) | 712 | 89 | 89 | 890 |
| | CHANGE_UNARY_OPERATOR (D) | 511 | 64 | 63 | 638 |
| | DELETE_THROWS_EXCEPTION (G) | 328 | 41 | 41 | 410 |
| | SWAP_ARGUMENTS (H) | 304 | 38 | 37 | 379 |
| | ADD_THROWS_EXCEPTION (F) | 98 | 12 | 12 | 122 |

Table 5: Statistics of ManySStuBs4J benchmark.

| | Error Type | Train | Valid | Test | All |
|---|---|---|---|---|---|
| High-resource | SINGLE_STMT | 24,346 | 3,044 | 3,043 | 30,433 |
| | CHANGE_STRING_LITERAL | 14,285 | 1,786 | 1,785 | 17,856 |
| | CHANGE_IDENTIFIER_USED | 5,035 | 630 | 629 | 6,294 |
| | CHANGE_BINARY_OPERAND | 3,434 | 430 | 429 | 4,293 |
| | SAME_FUNCTION_MORE_ARGS | 3,061 | 383 | 383 | 3,827 |
| | WRONG_FUNCTION_NAME | 2,813 | 352 | 351 | 3,516 |
| | CHANGE_NUMERIC_LITERAL | 2,480 | 310 | 310 | 3,100 |
| | ADD_FUNCTION_AROUND_EXPRESSION | 2,318 | 290 | 290 | 2,898 |
| | CHANGE_ATTRIBUTE_USED | 2,206 | 276 | 276 | 2,758 |
| | SINGLE_TOKEN | 1,895 | 237 | 237 | 2,369 |
| | ADD_METHOD_CALL | 1,290 | 161 | 161 | 1,612 |
| | MORE_SPECIFIC_IF | 1,101 | 138 | 137 | 1,376 |
| | ADD_ELEMENTS_TO_ITERABLE | 1,070 | 134 | 133 | 1,337 |
| | SAME_FUNCTION_LESS_ARGS | 1,050 | 131 | 131 | 1,312 |
| Low-resource | CHANGE_BOOLEAN_LITERAL | 907 | 114 | 113 | 1,134 |
| | ADD_ATTRIBUTE_ACCESS | 716 | 90 | 89 | 895 |
| | CHANGE_BINARY_OPERATOR | 681 | 85 | 85 | 851 |
| | SAME_FUNCTION_WRONG_CALLER | 558 | 70 | 70 | 698 |
| | CHANGE_KEYWORD_ARGUMENT_USED | 470 | 59 | 59 | 588 |
| | LESS_SPECIFIC_IF | 382 | 48 | 48 | 478 |
| | CHANGE_UNARY_OPERATOR | 318 | 40 | 40 | 398 |
| | SAME_FUNCTION_SWAP_ARGS | 150 | 18 | 19 | 187 |
| | CHANGE_CONSTANT_TYPE | 118 | 15 | 15 | 148 |

Table 6: Statistics of TSSB benchmark.

| | Error Type | Train | Valid | Test | All |
|---|---|---|---|---|---|
| High-resource | no-invalid-this | 13,101 | 1,456 | 1,609 | 16,166 |
| | no-undef | 8,614 | 958 | 1,064 | 10,636 |
| | no-unused-vars | 6,289 | 699 | 777 | 7,765 |
| | comma-style | 5,180 | 576 | 639 | 6,395 |
| | no-redeclare | 5,167 | 575 | 639 | 6,381 |
| | no-extra-semi | 4,834 | 537 | 598 | 5,969 |
| | no-unreachable | 3,826 | 426 | 473 | 4,725 |
| | prefer-rest-params | 3,675 | 405 | 454 | 4,534 |
| | no-debugger | 3,372 | 375 | 417 | 4,164 |
| | no-throw-literal | 3,300 | 367 | 408 | 4,075 |
| | guard-for-in | 2,616 | 291 | 324 | 3,231 |
| | no-console | 2,484 | 276 | 307 | 3,067 |
| | no-useless-escape | 2,364 | 263 | 293 | 2,920 |
| | prefer-spread | 2,001 | 221 | 244 | 2,466 |
| | no-dupe-keys | 1,765 | 197 | 219 | 2,181 |
| | no-empty | 1,665 | 184 | 206 | 2,055 |
| | no-process-exit | 1,225 | 137 | 152 | 1,514 |
| | no-cond-assign | 1,194 | 132 | 146 | 1,472 |
| | no-extra-boolean-cast | 1,180 | 132 | 146 | 1,458 |
| | generator-star-spacing | 1,130 | 126 | 140 | 1,396 |
| | no-constant-condition | 1,016 | 112 | 123 | 1,251 |
| Low-resource | no-array-constructor | 793 | 89 | 98 | 980 |
| | no-inner-declarations | 671 | 75 | 84 | 830 |
| | no-fallthrough | 601 | 67 | 75 | 743 |
| | no-case-declarations | 584 | 66 | 73 | 723 |
| | no-extra-bind | 547 | 59 | 68 | 674 |
| | no-self-assign | 494 | 55 | 61 | 610 |
| | valid-typeof | 436 | 49 | 54 | 539 |
| | constructor-super | 375 | 42 | 47 | 464 |
| | no-new-object | 360 | 41 | 45 | 446 |
| | no-caller | 360 | 41 | 45 | 446 |
| | no-extend-native | 358 | 40 | 45 | 443 |
| | require-yield | 347 | 39 | 43 | 429 |
| | no-unsafe-negation | 342 | 38 | 43 | 423 |
| | no-this-before-super | 333 | 38 | 42 | 413 |
| | no-new-wrappers | 291 | 33 | 36 | 360 |
| | no-global-assign | 257 | 29 | 32 | 318 |
| | no-const-assign | 224 | 25 | 28 | 277 |
| | no-sparse-arrays | 191 | 22 | 24 | 237 |
| | getter-return | 163 | 19 | 21 | 203 |
| | no-duplicate-case | 157 | 18 | 20 | 195 |
| | no-unused-labels | 151 | 17 | 19 | 187 |
| | no-empty-pattern | 144 | 16 | 18 | 178 |
| | no-func-assign | 118 | 14 | 15 | 147 |
| | no-dupe-class-members | 94 | 11 | 12 | 117 |
| | no-class-assign | 89 | 10 | 12 | 111 |
| | use-isnan | 56 | 7 | 8 | 71 |
| | no-unsafe-finally | 50 | 6 | 7 | 63 |
| | for-direction | 40 | 5 | 5 | 50 |
| | no-ex-assign | 32 | 4 | 4 | 40 |
| | no-compare-neg-zero | 9 | 2 | 2 | 13 |
| | no-new-symbol | 18 | 1 | 1 | 10 |

Table 7: Statistics of TFix benchmark.