# OpenReview forum: "Towards Low-Resource Automatic Program Repair with Meta-Learning and Pretrained Language Models"
_EMNLP/2023/Conference — EMNLP 2023 Main_

### Official Review · Reviewer_A13G · 2023-07-27

**Soundness:** 4

**Excitement:**

4: Strong: This paper deepens the understanding of some phenomenon or lowers the barriers to an existing research direction.

**Paper Topic And Main Contributions:**

This paper formulates automatic program repair (APR) for low-resource error types and proposes a meta-learning framework to address this. More concretely, Meta-APR is a CodeT5-based model which is first meta-trained on high-resource bug-fix pairs using the Reptile algorithm and then fine-tuned on low-resource error types, in 0-shot, 10-shot, 50-shot, and 100-shot settings. For training and evaluation, three existing datasets with labeled error types (TFix-JavaScript, ManySStuBs4J-Java, and TSSB-M-Python) are curated, such that any error type with less than 1,000 examples is considered low-resource while all others are considered high-resource. Meta-APR is evaluated against baselines in low-resource finetuning, transfer learning, and multi-task learning settings, as well as an alternative meta-learning baseline. Overall, their approach outperforms baselines. Moreover, they conduct a small-scale study against ChatGPT, and they find that their approach outperforms 0-shot and 1-shot ChatGPT for low-resource APR.

**Questions For The Authors:**

A) My understanding is that in Reptile, SGD is performed on each task one at a time. That is, the "Update error-specific APR parameters with gradient" step in Algorithm 1 would correspond to a specific error type. However, it seems that the batches that are used are randomly segmented from all high-resource error types (Line 284-286), suggesting that each batch could have examples corresponding to many different error types. If this is the case, then which "error-specific" parameters are updated?

B) In Lines 112-116, the authors claim that their approach is model-agnostic and generalizable to UniXcoder. In Tables 2-4, I see that is the case when comparing to the low-resource and transfer-learning settings. However, the multi-task learning results are not included for UniXcoder. Could you comment on how the multi-task learning performance compares to the meta-learning setting for UniXcoder?

C) In Section 5.3, the authors claim that ChatGPT performs poorly on APR for low-resource error types. However, it is not clear whether this is specific to low-resource cases or whether ChatGPT is performing poorly with respect to APR in general. Could you comment on how ChatGPT (and Meta-APR) perform with respect to high-resource error types? (Also, it would be nice to see how Meta-APR compares to the other baselines in Tables 2-4 on these high-resource cases.)

**Reasons To Accept:**

- The authors formulate a novel task relating to low-resource automatic program repair (APR) and curate datasets that could be useful to the research community in further studying this task.
- Leveraging meta-learning for this task is quite clever in my opinion. By demonstrating the advantage of meta-learning over simple finetuning, transfer learning, and multi-task learning, the authors also empirically highlight the value of this approach.
- The authors conduct very thorough experiments with many different baselines across different learning settings, and they even compare to ChatGPT at a small-scale to demonstrate how their approach stacks up against some of the most recent LLMs.
- The authors conduct extensive analysis of their results, with respect to length and also the distribution of the latent space. Figure 5 is especially useful in demonstrating how meta-learning helps align representations close together.

**Reasons To Reject:**

- It is unclear to me whether the results in this work will generalize to more realistic settings with software bugs "in the wild." Namely, the authors assume that the error type is labeled, and in fact, this is part of the input to the model (Line 350). In real-world settings, such a label will likely not be available. While one could argue that a classifier could be trained for determining the error type, that may also not be possible in realistic settings in which you have very complex and diverse bugs, unlike the single-statement and syntax/stylistic errors that are considered in this work. Additionally, it seems that one would have to finetune the model separately for each low-resource error type. This is not that feasible in practice, especially as the scale of models increases and the number of error types increase.
- The meta-training algorithm is a bit confusing and not explained clearly. This is one of the main contributions of this work, so this should be further clarified. I hope the authors can answer Question A below during the rebuttal period to provide further clarity on this.

**Reproducibility:**

5: Could easily reproduce the results.

**Reviewer Confidence:**

4: Quite sure. I tried to check the important points carefully. It's unlikely, though conceivable, that I missed something that should affect my ratings.

**Typos Grammar Style And Presentation Improvements:**

- APR parameters -> model parameters
- Explain Reptile in more detail since that is the backbone of your approach.
- Explain the MAML baseline more clearly and highlight the major differences.

---

> ### Author Rebuttal · Authors · 2023-08-29
>
> We thank the reviewer for the appreciation and support for our work. We are glad that you find our formulated low-resource APR task novel, and our curated dataset useful to the research community in further studying this task. We are further encouraged that you champion our clever design to leverage meta-learning for low-resource APR tasks, and find our work both technically and empirically sound, and backed with very thorough experiments and extensive analysis.
>
> Below we make every effort to address your questions. If you believe that our rebuttal is satisfactory, it would be great if you could consider increasing your soundness score.
>
> # Q1: “It is unclear to me whether the results in this work will generalize to more realistic settings with software bugs "in the wild." Namely, the authors assume that the error type is labeled, and in fact, this is part of the input to the model (Line 350). In real-world settings, such a label will likely not be available. While one could argue that a classifier could be trained for determining the error type, that may also not be possible in realistic settings in which you have very complex and diverse bugs. ”
>
> We appreciate your concerns and insights, which have prompted us to provide a clearer perspective on the versatility and applicability of our approach.
>
> The rationale behind our use of error types stems from our aim to delineate the distinctions between high-resource and low-resource bug scenarios. Regarding the concern about the availability of error type labels in real-world settings, we want to clarify our approach's versatility. While our study demonstrates the feasibility of obtaining error types from various sources, including static rules, analyzers, and interpreters/compilers, we acknowledge that complex and diverse bugs may not always fit neatly into predefined categories. We want to emphasize that the categorization of error type is just one facet of our approach.
>
> We believe that the underlying principles of our method are adaptable to other forms of categorization, such as project-specific attributes or patterns, thereby increasing the applicability of our approach across diverse scenarios. In our evaluation, we have showcased the effectiveness of our approach across multiple APR datasets in 3 programming languages with different definitions of error types. This reinforces the potential generalizability of this approach in addressing more complex real-world bug scenarios by embracing different categorization strategies and data sources.
>
> # Q2: “Additionally, it seems that one would have to finetune the model separately for each low-resource error type. This is not that feasible in practice, especially as the scale of models increases and the number of error types increase.”
>
> We did NOT have to finetune the model separately for each low-resource error type. Across our methods and all the baselines, we always combine the low-resource data with different error types together for joint training. We just need to inject the error information into the input.
>
> # Q3: “The meta-training algorithm is a bit confusing and not explained clearly…it seems that the batches that are used are randomly segmented from all high-resource error types (Line 284-286), suggesting that each batch could have examples corresponding to many different error types. If this is the case, then which "error-specific" parameters are updated?”
>
> We will provide more clarifications in the revised paper to address this. Your understanding of batches being randomly segmented from all high-resource error types is correct. For the “error-specific” parameters, they are error-specific in a sense that the error type information is embedded in the input, and the meta-learning algorithm implicitly captures this categorization information when updating the model parameters. In summary, our method design does not require updating the parameters for each type in a batch separately but models multiple types jointly, thereby being more efficient.
>
> # Q4: “Could you comment on how the multi-task learning performance compares to the meta-learning setting for UniXcoder?”
>
> Thanks for pointing this out. We did indeed assess UniXcoder in multi-task learning and observed its consistent inferior performance compared to CodeT5, echoing findings from other learning scenarios (see finetuning-only and transfer learning in Tables 2,3 and 4). For UniXcoder’s performance across different learning settings, we found that multi-task learning with upsampling is mostly comparable to the transfer learning baseline and consistently worse than meta-learning across 3 benchmarks. We will incorporate these results back for a more comprehensive paper.
>
> # Q5: “In Section 5.3, the authors claim that ChatGPT performs poorly on APR for low-resource error types. However, it is not clear whether this is specific to low-resource cases or whether ChatGPT is performing poorly with respect to APR in general. Could you comment on how ChatGPT (and Meta-APR) perform with respect to high-resource error types?”
>
> We appreciate the suggestion to evaluate our methods and other baselines like ChatGPT on high-resource error types. We will evaluate ChatGPT on high-resource error types to provide clarification on this. We want to emphasize two findings of ChatGPT’s common behaviour in the context of APR: 1) ChatGPT often predicts “no bugs” as it might require more semantic information to understand the context; 2)  ChatGPT often performs pretty well in fixing bugs related to syntax errors.
>
> From our preliminary evaluation of Meta-APR on high-resource bugs, we observed that both Meta-APR and other baselines such as multi-task learning can achieve good results with comparable performance, indicating that both methods can perform reasonably well when there is abundant training data. However, we want to emphasize that our approach is rooted in the goal of addressing low-resource APR tasks effectively, where data scarcity is a key challenge. The primary objective of our work is to excel in low-resource conditions, where traditional approaches may struggle due to a lack of sufficient training data. From our comprehensive evaluation (Table 2,3,4), we have showcased that our method's efficacy becomes increasingly significant as the data scarcity intensifies.
>
> # Q6: Presentation improvement: more explanations on Reptile and MAML
> Thanks for the suggestion! We will elaborate these two meta-learning methods in our final version.

---

### Official Review · Reviewer_yvFS · 2023-08-03

**Soundness:** 3

**Excitement:**

3: Ambivalent: It has merits (e.g., it reports state-of-the-art results, the idea is nice), but there are key weaknesses (e.g., it describes incremental work), and it can significantly benefit from another round of revision. However, I won't object to accepting it if my co-reviewers champion it.

**Paper Topic And Main Contributions:**

This paper proposes to apply Reptile algorithm for meta-learning to low-resource automatic program repair (APR). They construct a low-resource training set by sampling 10, 50 and 100 datapoints for various error types across three APR benchmarks. Their results demonstrate that their method is effective with 10, 50, 100 samples per error type and achieves better performance than transfer/multitask learning. Their findings also show that the first-order (Reptile) algorithm outperforms the second-order algorithm (MAML).

**Questions For The Authors:**

1. In the meta-training process, error types are made visible to the model where it can learn to distinguish between types. Is this setting consistent with your baselines? How would your method work if this information were not available?

2. After fine-tuning the LLMs to carry out Automatic Program Repair, the programs provided are expected to contain one error, which is the default setting of APR LLMs. Have you tried the setting where ChatGPT was explicitly informed about this—"the following program contains one error"?

3. Since your low-resource definition is under "1000 samples", why not use the naturally-formed low-resource set: all samples as long as they don't exceed 1000? Why have you excluded UniXCoder and CodeBert's performances for multi-task learning?

Comments on rebuttal

The author's response clarified various issues that made it clear the baselines used are fair comparisons and addressed some of our concerns with the paper. However, the task still seems quite artificial where the data had to be artificially limited to illustrate the advantages of the system, and the paper does not introduce any new algorithms or datasets, so I changed the scores from 2,2 to 3,3

**Reasons To Accept:**

1. The paper innovatively draws attention to the largely unexplored area of low-resource automatic program repair, presenting criteria to form low-resource datasets.

2. The authors skillfully utilize the meta-learning method Reptile Algorithm for fine-tuning large language models (LLMs) on high-resource automatic program repair tasks. This enables efficient adaptation to low-resource tasks with a range of 10 to 100 examples.

3. APR is an interesting and important problem and exploring how LLMs can help address it is timely and significant.

**Reasons To Reject:**

1. The study's design is somewhat artificial. The authors artificially restrict the number of data points for tasks with more than 100 data points, creating an arguably unrealistic low-resource scenario. For the application of their task, the bug type must be discernible during both the fine-tuning/training and testing phases, deviating substantially from real-world conditions. Testing on a benchmark where the available data is naturally limited, rather than artificially limiting the available training data, would provide better motivation for the method to address a real-world problem.

2. While the authors' method proves effective for tasks with minimal data, the improvements over multi-task transfer learning on 100 samples are minor, potentially curbing its efficacy for bug types with larger sample sizes.

3. The experiments are not sufficient. The lack of exposure to high-resource data for low-resource fine-tuning makes the comparison less balanced. The paper does not present performance outcomes for UniXCoder under a multi-task and upsampling scenario. Additionally, the authors only use the Exact Match metric for evaluating the generated repair, which overlooks the potential correctness of fixes that do not exactly match. When available, the authors should also present the Error Removal results.

4. The novelty of the work is somewhat limited as it primarily relies on applying the existing Reptile algorithm to the APR task; there are no new algorithms presented.

**Reproducibility:**

4: Could mostly reproduce the results, but there may be some variation because of sample variance or minor variations in their interpretation of the protocol or method.

**Reviewer Confidence:**

4: Quite sure. I tried to check the important points carefully. It's unlikely, though conceivable, that I missed something that should affect my ratings.

---

> ### Author Rebuttal · Authors · 2023-08-29
>
> Thank you for the constructive comments. We are glad that you acknowledge our novelty to draw attention to the largely unexplored area of low-resource APR, and find our approach of combining LLMs with a meta-learning framework skillful and efficient. We are further encouraged that you recognize our work as a timely and significant contribution to explore LLMs for an interesting and important task of APR.
>
> It seems that your concerns on soundness derive from the study’s design. We will make every effort to address these points and provide clarification in our revised manuscript. If your confusion or concern is addressed, we request your kind support by changing your stance.
>
> # Q1: “The study's design is somewhat artificial. The authors artificially restrict the number of data points for tasks with more than 100 data points, creating an arguably unrealistic low-resource scenario. For the application of their task, the bug type must be discernible during both the fine-tuning/training and testing phases, deviating substantially from real-world conditions. Testing on a benchmark where the available data is naturally limited, rather than artificially limiting the available training data, would provide better motivation for the method to address a real-world problem.”
>
> We appreciate your feedback regarding the study's design. We acknowledge your concerns about the potential artificial nature of our study and would like to provide further clarification to elaborate on the rationale behind our methodology. We will add more discussion on the applicability and generalizability of our approach in our revised version.
>
> ## Rationale behind the formulation of low-resource APR by controlling the number of data points
> As you aptly noted, our work seeks to address the novel problem of low-resource APR tasks. In this pioneering endeavour, we faced a significant challenge: the absence of existing datasets that accurately capture the intricacies of this particular problem domain. Given this limitation, we embarked on a methodical exploration of available datasets to establish a foundation for our study.
>
> Our decision to create artificial low-resource scenarios by restricting the number of data points for tasks was born out of necessity. This approach allowed us to simulate low-resource conditions and evaluate our proposed method's effectiveness in situations where data is scarce at different levels (i.e. 10/50/100 instances)—a crucial aspect of our research objective. By focusing on error types and strategically formulating these scenarios, we aimed to capture the essence of low-resource APR tasks while working within the constraints of the data at hand.
>
> Although we acknowledge the potential artificiality introduced by this approach, we believed that this was a pragmatic way to initiate the study of low-resource APR tasks in the absence of suitable datasets. Our intention was to spark discussion and exploration in a relatively uncharted research area, paving the way for future investigations that can refine and expand upon our initial findings. We are encouraged to see an explicit recognition on this point from Reviewer A13G that our curated datasets could be useful to the research community in further studying this task.
>
> ## Rationale behind the use of error types
> The rationale behind our use of error types stems from our aim to delineate the distinctions between high-resource and low-resource bug scenarios. We agree that relying solely on error types might introduce limitations, but it is essential to emphasize that this categorization is just one facet of our approach. We firmly believe that the underlying principles of our method are adaptable to other forms of categorization, such as project domains, thereby increasing the applicability of our approach across diverse scenarios.
>
> Additionally, while error types are not always readily available, we want to highlight that our study was designed to demonstrate the feasibility of obtaining error types from various sources, including static rules, analyzers, dynamic interpreters, and compilers.  In our work, we have demonstrated the effectiveness of our approach across multiple APR datasets in 3 programming languages (PLs) with different definitions of error types. While the availability of error types might vary, we believe that our work underscores the potential utility and generalizability of this approach in addressing low-resource scenarios across various programming contexts, thus mitigating the concern about over-reliance on specific error types and data sources.
>
> # Q2: “While the authors' method proves effective for tasks with minimal data, the improvements over multi-task transfer learning on 100 samples are minor, potentially curbing its efficacy for bug types with larger sample sizes”
>
> Our approach is rooted in the goal of addressing low-resource APR tasks effectively, where data scarcity is a key challenge. We acknowledge your observation that, while our method demonstrates strong performance in scenarios with minimal data, the gains observed over multi-task transfer learning on 100 samples may appear relatively minor. We appreciate your attention to this aspect, and we would like to offer the following insights to provide a more comprehensive understanding of our approach:
>
> * **Focus on Low-Resource Scenarios:** The primary objective of our work is to excel in low-resource conditions, where traditional approaches may struggle due to a lack of sufficient training data. While the improvements in scenarios with 100 samples might appear less pronounced, it's important to note that our method's efficacy becomes increasingly significant as the data scarcity intensifies.
>
> * **Consistent Gains across Multiple APR Tasks:** First, the minor improvements observed in tasks with 100 samples should not overshadow the substantial gains our method demonstrates when operating under extremely low-resource scenarios, such as those with fewer than 50 samples. These scenarios are where our method truly shines, offering solutions in situations where other methods might fail. Besides, we have demonstrated consistent performance gains in low-resource scenarios across various programming contexts and observed similar performance patterns across different numbers of data samples. We believe these findings would be insightful for the research community to further explore this problem.
>
> # Q3: “The experiments are not sufficient. The lack of exposure to high-resource data for low-resource fine-tuning makes the comparison less balanced. The paper does not present performance outcomes for UniXCoder under a multi-task and upsampling scenario. Additionally, the authors only use the Exact Match metric for evaluating the generated repair, which overlooks the potential correctness of fixes that do not exactly match. When available, the authors should also present the Error Removal results.”
>
> Your observations highlight crucial aspects that warrant further clarification and enhancement, and we would like to address each of these concerns comprehensively.
>
> ## The lack of exposure to high-resource data for low-resource fine-tuning:
> We would like to clarify that we did, in fact, conduct the experiment you mentioned. Our methodology involved a comprehensive evaluation under various scenarios, including exposure to high-resource data during the fine-tuning phase in different strategies (see transfer learning and multi-task learning w/ or w/o upsampling in Tables 2,3, and 4). We understand the importance of a balanced comparison, and therefore, this experiment was an integral part of our research design.
>
> ## The paper does not present performance outcomes for UniXcoder under a multi-task and upsampling scenario.
>  We did indeed assess UniXcoder in multi-task learning and observed its consistent inferior performance compared to CodeT5, echoing findings from other learning scenarios (see finetuning-only and transfer learning in Table 2,3,4). We omitted these results for clarity in presentation. We value your suggestion and will incorporate these results back for a more comprehensive paper.
>
> ## Limitations of using Exact Match metric
> We recognize your concern regarding the limitations of Exact Match. Your suggestion to include Error Removal results is indeed valuable. We plan to explore the Error Removal metric to make our evaluation more comprehensive. For Exact Match, we note that while is a kind of too strict metric that necessitates that fixes precisely match the ground-truth developer-written fixes, it is commonly utilized to uphold correctness standards, especially in cases where static analyzers or unit tests are not available. From the TFix paper, we also find that the Exact Match metric aligns well with the Error Removal metric, providing further validation to the soundness of our results based on the Exact Match metric.
>
> # Q4: The novelty of the work is somewhat limited as it primarily relies on applying the existing Reptile algorithm to the APR task; there are no new algorithms presented.
>
> Your observation about our reliance on the existing Reptile algorithm is valid. We want to take the opportunity to provide a comprehensive understanding of our contributions.
>
> Our novelty indeed extends beyond merely applying the Reptile algorithm. We take pride in formulating a novel task within the realm of APR: the challenge of low-resource APR. This formulation presents a distinctive and meaningful contribution to the field (acknowledged by both you and Reviewer A13G). It entails curating datasets that not only serve our research but also have the potential to benefit the broader research community in further exploring this novel task (acknowledged by Reviewer A13G). These acknowledgments reinforce our belief that our approach is both innovative and valuable. Leveraging Reptile meta-learning with strategic adaptation to address this task presents another contribution of our work. Our focus is on exploring the potential of established methods in novel contexts, rather than presenting entirely new algorithms.
>
> # Q5: In the meta-training process, error types are made visible to the model where it can learn to distinguish between types. Is this setting consistent with your baselines? How would your method work if this information were not available?
>
> Yes, the input format (see Line 350) including error types are consistent with all baselines and our methods. We follow the TFix work to employ such input formats and aim to explore how different training methods affect the knowledge transfer from fixing high-resource bugs to low-resource bugs.
>
> While we did not explicitly explore an ablated setting where error type information is absent, we appreciate your suggestion which prompts us to further consider whether the model itself can potentially implicitly cluster error types. We will explore this setting and incorporate more ablation results to our revised paper.
>
> # Q6: After fine-tuning the LLMs to carry out Automatic Program Repair, the programs provided are expected to contain one error, which is the default setting of APR LLMs. Have you tried the setting where ChatGPT was explicitly informed about this—"the following program contains one error"?
>
> In our prompt to ChatGPT, we explicitly provide the buggy line and ask the ChatGPT to fix the buggy line. A brief example can be found in Figure 6.
>
> # Q7: Since your low-resource definition is under "1000 samples", why not use the naturally-formed low-resource set: all samples as long as they don't exceed 1000?
>
> We appreciate your insightful suggestion to include a setting of a naturally-formed low-resource set. We want to provide clarification on our choice.
>
> Our decision to use different instance thresholds (10/50/100) for creating low-resource scenarios across multiple datasets is rooted in the desire to control the degree of low-resource more accurately. As we include 3 different datasets, it doesn't account for the varying data distributions within each dataset if directly using the low-resource set cut by the "1000 samples" threshold. By defining the low-resource scenarios based on different instance thresholds, we aim to maintain a consistent level of data scarcity across datasets with distinct characteristics.

---

### Official Review · Reviewer_rer7 · 2023-08-04

**Soundness:** 4

**Excitement:**

4: Strong: This paper deepens the understanding of some phenomenon or lowers the barriers to an existing research direction.

**Paper Topic And Main Contributions:**

**Summary:**

The paper investigates low-resource automatic program repair (APR) based on the different types of error. It mainly shows that different types of software bugs have an imbalanced distribution, and the APR models often only capture the frequent patterns (error types), and have difficulty to deal with low-resource error types. To deal with this issue, the paper proposes a meta-learning framework to repair the low-resource software bugs. This Meta-APR framework first learns the knowledge from high-resource bugs, and then fine-tuned on the target low-resource bugs.

**Contributions:**

- The paper investigates the pre-trained code models’ capabilities in dealing with low-resource automatic program repair (APR) tasks.
- The paper proposes a meta-learning framework to generate fixes for low-resource bugs with limited training data.
- The provided results show that the meta learning framework consistently enhances the performance of the model in low-resource APR tasks.
- Furthermore, the paper shows that the proposed model perform better than large close source models such as ChatGPT.

**Questions For The Authors:**

- In Fig 1, what do high and low resources represent? I think it is a bit confusing, as the high-resource is on top of the meta-learning framework. And the right part contains both high and low resource data.
- Could you please explain in more detail how you “find that the bugs successfully fixed by Meta-APR tend to be shorter than the ones that are incorrectly fixed.” from Fig 3. a?
- In 4.2, the details of the baselines are provided. However, it is not clear what exactly these baselines are. In Particular, it is not clear what exactly transfer-learning and multi-task learning are.
- I appreciate that you also considered ChatGPT as the baseline. Is there any particular reason that you did not check the results with two or more shots (examples)?

**Reasons To Accept:**

- The paper investigates the pre-trained code models’ capabilities in dealing with low-resource automatic program repair (APR) tasks. - Furthermore, it proposes to use a meta-learning framework to deal with these tasks.
It provides detailed evaluation results to show the effectiveness of the meta-learning framework in dealing with low-resource APR tasks.

**Reasons To Reject:**

- There is not a major novelty in the meta-learning framework (However, I think that is a minor issue).
- There are some presentation issues in the paper, including unclarities in some of the Figs.
- Some of the baseline details are not clear (Please see “Questions For the Authors”).

**Reproducibility:**

3: Could reproduce the results with some difficulty. The settings of parameters are underspecified or subjectively determined; the training/evaluation data are not widely available.

**Reviewer Confidence:**

3: Pretty sure, but there's a chance I missed something. Although I have a good feel for this area in general, I did not carefully check the paper's details, e.g., the math, experimental design, or novelty.

---

> ### Author Rebuttal · Authors · 2023-08-29
>
> We thank the reviewer for the appreciation and support for our work. We are glad that you recognize our motivation of investigating the pre-trained code models’ capabilities within a meta-learning framework in dealing with low-resource APR tasks, and find our proposed approach technically sound and backed with detailed evaluation results.
>
> We address your concerns as follows:
> # Q1: “In Fig 1, what do high and low resources represent? I think it is a bit confusing, as the high-resource is on top of the meta-learning framework. And the right part contains both high and low resource data..”
> Thanks for flagging this. In Figure 1, the high-resource one refers to the more abundant and common error types that are used to meta-train the backbone model, while the low-resource ones represent the more rare error types that our model adapts into.
>
> Indeed the right part of the figure contains both high (green circles) and low resource (triangles) test data, which is meant to highlight their different distributions. Specifically, the target low-resource bugs are highlighted with ellipses and arrows to represent the transferability from high-resource to low-resource scenarios. We will make it more clear in our revised version.
>
> # Q2: “Could you please explain in more detail how you “find that the bugs successfully fixed by Meta-APR tend to be shorter than the ones that are incorrectly fixed.” from Fig 3. a?”
>
> The plot in Fig 3 (a) presents a cumulative fraction of programs by the number of tokens in the source buggy patch, grouped by whether Meta-APR can correctly fix it or not: the blue line for the correct fixes and the orange one for the wrong fixes. This aims to showcase the distribution differences of the buggy patches between which Meta-APR can and cannot fix.
>
> As illustrated in Fig 3 (a), we can observe that the blue line is consistently above the orange one, and if we select a fixed cumulative fraction (say 0.5 in the y-axis), the blue line (correct fixes) will have fewer number of tokens (i.e. shorter) than the orange one (wrong fixes).
>
> # Q3: “In 4.2, the details of the baselines are provided. However, it is not clear what exactly these baselines are. In Particular, it is not clear what exactly transfer-learning and multi-task learning are..”
> Thanks for bringing this to our attention. Let us further explain these baselines here and we will make it clearer in our revised version.
> Transfer-learning from high-resource to low-resource bugs refers to that we first finetune the model on the high-resource training data, and then have another round of finetuning on the low-resource training data.
> Multi-task learning on both high-resource and low-resource bugs refers to that we mix these both data and have only one round of finetuning. Under the multi-task learning setting, we further explore ablations of whether to upsampling the low-resource bugs at a similar size of the high-resource ones.
>
> After the training, the model will be evaluated on the low-resource test data to examine its low-resource APR capabilities. By considering all these training strategies for comparison, we aim to demonstrate the consistent benefits of our meta-learning strategy.
>
> # Q4: “I appreciate that you also considered ChatGPT as the baseline. Is there any particular reason that you did not check the results with two or more shots (examples).”
>
> Thanks for your appreciation. We indeed explored using two and three examples for the few-shot in-context learning with ChatGPT. Specifically, we randomly selected bug-fix examples from the same error type and concatenate them as the prefix prompt for guidance. However, we did not observe significant improvements compared to the one-shot prompt. Besides, we had a similar observation that few-shot examples are not always beneficial for low-resource APR and might introduce some noises compared to zero-shot settings. We will add more discussion in the final version.

---

### Meta-Review · Area_Chair_xWDe · 2023-09-01

**Recommendation:** Accept (Poster)
**Confidence:** 4

**Metareview:**

The authors have contributed to an important and challenging area of program repair, and have raised the important issue of the lack of data in the area. Their evaluations create an artificial scarcity of data which can be critiqued, but their evaluations demonstrating the value of their method are comprehensive. Some writing and figure captions can be clarified to improve presentation and readability.

---

### Decision · Program_Chairs · 2023-10-07

**Decision:**

Accept-Main

**Comment:**

The authors have contributed to an important and challenging area of program repair, and have raised the important issue of the lack of data in the area. Their evaluations create an artificial scarcity of data which can be critiqued, but their evaluations demonstrating the value of their method are comprehensive. Some writing and figure captions can be clarified to improve presentation and readability.